# Hyperbolic and Mixed Geometry Graph Neural Network

**Xinyue Cui**                                                                XINYUECU@USC.EDU
*University of Southern California*

**Rishi Sonthalia**                                                    RSONTHAL@MATH.UCLA.EDU
*University of California, Los Angeles*

**Editors:** Sophia Sanborn, Christian Shewmake, Simone Azeglio, Arianna Di Bernardo, Nina Miolane

## Abstract

Hyperbolic Graph Neural Networks (GNNs) have shown great promise for modeling hierarchical and graph-structured data in the hyperbolic space, which reduces embedding distortion comparing to Euclidean space. However, existing hyperbolic GNNs implement most operations through differential and exponential maps in the tangent space, which is a Euclidean subspace. To avoid such complex transformations between the hyperbolic and Euclidean spaces, recent advances in hyperbolic learning have formalized hyperbolic neural networks based on the Lorentz model that realize their operations entirely in the hyperbolic space via Lorentz transformations Chen et al. (2022). Here, we adopt the hyperbolic framework from Chen et al. (2022) and propose a family of hyperbolic GNNs with greater modeling capabilities as opposed to existing hyperbolic GNNs. We also show that this framework allows us to have neural networks with both hyperbolic layers and Euclidean layers that can be trained jointly. Our experiments demonstrate that our fully hyperbolic GNNs lead to substantial improvement in comparison with their Euclidean counterparts.

## 1. Introduction

The geometry of the representations learned by a neural network has proved to be important. Recently, due to the realization that hierarchical data can be well represented in hyperbolic space (Hamann, 2017; Dyubina and Polterovich, 1999), many algorithms have been developed to embed data into hyperbolic space (Nickel and Kiela, 2017, 2018; Sonthalia and Gilbert, 2020; Sala et al., 2018) and many different hyperbolic neural network architectures have been proposed as well (see survey Peng et al. (2021)).

Given a manifold $\mathcal{M}$ whose geometry we want to use, the first issue with describing a neural network is that we need to define a linear function from $\mathcal{M}$ to itself. The standard method to do this is to use the tangent space. That is, we use the Logarithmic map from the manifold to the tangent space, which is Euclidean. We perform one layer of a neural network here and then we map back to the manifold using the Exponential map. However, this has many issues. First, we need to know the Logarithmic and Exponential maps. Second, even when these are known, they could be very computationally expensive to calculate or could have numerical stability issues. In the case of hyperbolic manifolds, we have additional numerical issues related to representing points on the manifold itself (Yu and Sa, 2019). Recent work (Chen et al., 2022) proposed a new method to get around this issue by parametrizing linear maps on Lorentz manifolds as maps on Euclidean space.

In this paper, we notice that the map defined in Chen et al. (2022) can be thought of a map from Euclidean space to Euclidean space. So instead of needing Riemannian

gradient descent, we can use standard gradient descent. This then allows us to seamlessly mix different Euclidean and hyperbolic layers. Hence we have created mixed geometry neural networks by replacing the linear layers by hyperbolic linear layers. We test the new architectures on standard graph classification datasets and show that the mixed geometry versions consistently outperform their Euclidean counterparts.

## 1.1. Related Work

Apart from Chen et al. (2022), there exists three prior papers on hyperbolic graph neural networks (Chami et al., 2019; Liu et al., 2019; Zhang et al., 2021). Each of these uses the older method of performing computations in the tangent space. See the survey Yang et al. (2022) for more details.

## 2. Hyperbolic Manifold

First, we briefly introduce the Hyperboloid or Lorentzian model of the hyperbolic manifolds. The Hyperboloid model $\mathbb{H}^k$ of the hyperbolic manifold is

$$\mathbb{H}^k = \left\{ x \in \mathbb{R}^{k+1} : x_0 > 0, x_0^2 - \sum_{i=1}^{k} x_i^2 = 1 \right\}.$$

Here distances are given via $d(x, y) = \operatorname{arccosh}\left( -\sum_{i=1}^{k} x_i y_i + x_0 y_0 \right)$. For the Hyperboloid model, we have explicit formulas for the exponential and logarithmic maps. Specifically, the exponential map is given by $\exp_x(v) = \cosh(\|v\|_{\mathbb{H}})x + \sinh(\|v\|_{\mathbb{H}})\dfrac{v}{\|v\|_{\mathbb{H}}}$, and the logarithmic map is given by $\log_x(y) = \dfrac{\operatorname{arccosh}(\beta)}{\sqrt{\beta^2 - 1}}(y - \beta x)$, where $\beta = -x_0 y_0 + \sum_{i=1}^{k} x_i y_i$ and $\|x\|_{\mathbb{H}} = -x_0^2 + \sum_{i=1}^{k} x_i^2$.

## 2.1. Using Tangent Space.

We are now ready to describe the standard hyperbolic graph neural network layer. Let $F$ be any graph neural network layer on Euclidean space, then the corresponding hyperbolic version is traditionally given as follows

$$HF(x) = \log(F(\exp(x)))$$

where exp, log are the Exponential and Logarithmic maps. In some cases, this can be simplified. However, in most cases this is computationally expensive.

## 2.2. Fully Hyperbolic Network.

The fully hyperbolic framework proposed in Chen et al. (2022) selected the Lorentz model as its feature space. The family of linear transformations between Lorentz models are called

Lorentz transformations, which can be decomposed into Lorentz rotation and Lorentz boost. The Lorentz rotation describes the rotation of spatial coordinates, while the Lorentz boost describes relative motion with constant velocity and without rotation of spatial axes.

**Definition 1** *The Lorentz rotation matrices are given by* $\mathbf{R} = \begin{pmatrix} 1 & \mathbf{0}^\top \\ \mathbf{0} & \tilde{\mathbf{R}} \end{pmatrix}$, *where* $\tilde{\mathbf{R}}^\top \tilde{\mathbf{R}} = \mathbf{I}$ *and* $\det(\tilde{\mathbf{R}}) = 1$.

**Definition 2** *Given a velocity* $\mathbf{v} \in \mathbb{R}^n, \|\mathbf{v}\| < 1$ *and* $\gamma = \frac{1}{\sqrt{1-\|\mathbf{v}\|^2}}$, *the Lorentz boost matrices are given by* $\mathbf{B} = \begin{pmatrix} \gamma & -\gamma\mathbf{v}^\top \\ -\gamma\mathbf{v} & \mathbf{I} + \frac{\gamma^2}{1+\gamma}\mathbf{v}\mathbf{v}^\top \end{pmatrix}$.

Both Lorentz boost and Lorentz rotation are defined on the Lorentz model and can be adopted as the basis for building fully hyperbolic neural networks.

In Lemma 2 in Chen et al. (2022), they show that maps that use the tangent space can model certain types of "pseudo-rotations" and cannot model Lorentz boosts. Hence they propose the following hyperbolic layer instead. Given a matrix $M = \begin{bmatrix} v \\ W \end{bmatrix}$ and an input $x$ of the hyperbolic linear layer, then the output is given by

$$z = \begin{bmatrix} \frac{\sqrt{\|Wx\|^2-1}}{v^T x} v \\ W \end{bmatrix} x.$$

They show that if $x$ lives in the hyperbolic manifold then so does $z$.

## 3. Mixed Geometry Networks

In this paper, we simplify the above layer to define a linear layer that maps $\mathbb{R}^d \to \mathbb{H}^k$. Here given a matrix $W \in \mathbb{R}^{k \times d}$ and a vector $x \in \mathbb{R}^d$, we define the output of this layer as follows

$$\hat{x} = Wx$$

and then the output $z$ is given by for $i > 0$, $z_i = \hat{x}_i$ and

$$z_0 = \sqrt{1 + \sum_{i=1}^{k} \hat{x}_i}.$$

Thus, here we can see that given any vector $x$, then $z$ lives in the Hyperboloid manifold. Here we write $z = HL(x; W)$.

We then define the mixed geometric hyperbolic version of three different types of graph neural network architectures. Specifically Graph Convolutional Neural Network (GCN) Kipf and Welling (2017), Graph Conv (GC) Morris et al. (2019), and Residual Gated Graph Conv (RGGC) Bresson and Laurent (2017).

### 3.1. GCN

A standard GCN layer is given by

$$X' = \hat{D}^{-1/2}\hat{A}\hat{D}^{-1/2}X\Theta,$$

where $A$ is adjacency matrix, $\hat{A} = A+I$ and $\hat{D}$ is the degree matrix. For the mixed geometry version, we modify the update rule to

$$X' = \hat{D}^{-1/2}\hat{A}\hat{D}^{-1/2}HL(X;\Theta).$$

### 3.2. GC

A standard GraphConv layer is given by

$$x_i' = W_1 x_i + W_2 \sum_{j \in \mathcal{N}(i)} e_{j,i} x_j,$$

where $e_{j,i}$ is the edge weight of the edge from $j$ to $i$. For the mixed geometry version, we modify the update rule to

$$x_i' = HL(x_i; W_1) + HL\left(\sum_{j \in \mathcal{N}(i)} e_{j,i} x_j; W_2\right).$$

### 3.3. RGGC

A standard Residual Gated Graph layer is given by

$$x_i' = W_1 x_i + W_2 \sum_{j \in \mathcal{N}(i)} \eta_{j,i} x_j,$$

where $\eta_{j,i} = \sigma(W_3 x_i + W_4 x_j)$. For the mixed geometry version, we modify the update rule to

$$x_i' = HL(x_i; W_1) + HL\left(\sum_{j \in \mathcal{N}(i)} \eta_{j,i} x_j.; W_2\right),$$

where $\eta_{j,i} = \sigma(HL(x_i; W_3) + HL(x_j; W_4))$.

As we can see we have defined a mixed geometry layer as we compute the addition in Euclidean space. This is different from Chen et al. (2022), which does the aggregation in hyperbolic space.

## 4. Experiments

To verify the effectiveness of the new method, we test the architecture on 3 different node classification datasets: Karate (Zachary, 1977), Cora (Yang et al., 2016), and PubMed (Yang et al., 2016). To do a fair comparison, all models have 2 layers, with 128 hidden dimensions. The models are trained for 10 epochs using Adam optimizer (Kingma and Ba,

2015). The datasets and Euclidean implementations were taken from Pytorch Geometric (Fey and Lenssen, 2019).

We trained each model for 10 times, and report the mean and standard deviation of test accuracy in Table 1. As we can see from the table, the hyperbolic version outperforms their Euclidean counterparts. Specifically, we see that not only does the mean accuracy increase, but the variance decreases as well.

We hypothesize that the change to loss landscape from the reparametrization is the reason for the improvement. In this way we postulate that Mixed Geometry Graph Neural Networks outperform standard graph neural networks only due to the change in the loss landscape. We also note that with our parametrization the symmetry between all of the neurons in a layer of a neural network have now been broken with one neuron in every layer being special.

| Model | Dataset | | |
|---|---|---|---|
| | Karate | Cora | PubMed |
| GCN | $0.46 \pm 0.08$ | $0.75 \pm 0.05$ | $0.74 \pm 0.01$ |
| MGCN | $0.47 \pm 0.07$ | $0.77 \pm 0.02$ | $0.75 \pm 0.02$ |
| GC | $0.53 \pm 0.06$ | $0.67 \pm 0.03$ | $0.62 \pm 0.06$ |
| MGC | $0.55 \pm 0.03$ | $0.78 \pm 0.02$ | $0.73 \pm 0.02$ |
| RGGC | $0.54 \pm 0.09$ | $0.68 \pm 0.04$ | $0.66 \pm 0.07$ |
| MRGGC | $0.57 \pm 0.04$ | $0.78 \pm 0.01$ | $0.73 \pm 0.02$ |

Table 1: Table comparing our models against their Euclidean counterparts on a variety of benchmark datasets in terms of test accuracy. All accuracies reported are averaged over 10 instances.

## 5. Future Work

While this provides initial evidence that this simple reparametrization can lead to improvements, there is still a lot of work to be done. In particular, this comparison is only on 3 different data sets, and one network design. A more comprehensive comparison on more datasets, more tasks (graph and edge classification), as well as comparison against prior hyperbolic graph neural network architectures are needed.

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
