# OpenReview forum: "Hyperbolic and Mixed Geometry Graph Neural Networks"
_NeurIPS.cc/2022/Workshop/NeurReps — NeurReps 2022 Poster_

### Official Review · Reviewer_wcVZ · 2022-10-10
**Interesting core idea but much more to be done on both theory and experiments**

**Confidence:** 4
**Soundness:** 2
**Presentation:** 3
**Contribution:** 2
**Overall Rating:** 5

**Summary:**

The paper proposes to extend commonly used GNN architectures to node representations living in the hyperbolic space by projecting the node-wise channel-mixing operation onto the hyperboloid model.

**Questions:**

See Weakness section above.

**Limitations:**

Yes.

**Recommended Decision:**

2: Borderline

**Relevance:**

3: Solid fit

**Strengths And Weaknesses:**

Strengts: The main idea of the paper consists in replacing the standard way of approaching hyperbolic GNN where operations are performed in the tangent space via the exponential map and then updates are mapped back to the manifold through the logarithmic map with a simple projection of Euclidean vectors directly onto the hyperboloid model. In its simplicity, this idea is worth investigating in the context of GNNs.

Weakness: This builds on previous works and the extensions so far are a bit limited. From a theoretical point of view it would be nice to explore if -- as partly hinted at -- the improvement derived from this extra projection step can be proven to be equivalent to a change in the loss so that the hyperboloid projection boils down to a regularization effect and perhaps no real hierarchical properties are being captured. The evaluation is also very limited. It'd be interesting for example to see if -- maybe as one would expect -- there is no improvement on datasets that do not possess a hierarchical structure.

**Submission Track:**

Extended Abstract (4 Page)

---

### Official Review · Reviewer_hZki · 2022-10-15
**Hyperbolic and Mixed Geometry Graph Neural Network**

**Confidence:** 4
**Soundness:** 3
**Presentation:** 3
**Contribution:** 3
**Overall Rating:** 8

**Summary:**

The authors propose the Mixed Geometry neural networks based on Lorentz transformations of the hyperbolic space. This framework allows to combine hyperbolic layers with Euclidean layers. The Mixed Geometry neural networks show better performance than classical ones on several node classification datasets.

**Questions:**

p.1 l.-17: The expression "a linear function from M to itself" is not rigorous since linear functions are only defined between vector spaces.

p.1 l.-15: euclidean -> Euclidean (capital letter)

p.1 l.-10: a word may be missing.

p.1 l.-6: Euclidean, Euclidean, Riemannian (capital letters)

p.1 l.-4 and l.-1: Euclidean

p.2 l.8: n should be k and n+1 should be 0

p.3 l.8: What do you mean by "linear layer"? The map x -> z is not linear (and H^k is not a vector space).

p.3 l.-8: missing hat on D?

p.3 l.-6: Please define curly N.

p.4 l.2 and l.9: Euclidean

**Limitations:**

The authors mention several limitations in Section 5, which is appreciated.

**Recommended Decision:**

3: Accept

**Relevance:**

4: Highly relevant

**Strengths And Weaknesses:**

The novelty is the new linear layer replacing the hyperbolic layer introduced in Chen et al., which is promising since it allows to combine linear and hyperbolic layers.

The paper is well written and the formulae on the hyperboloid look correct.

I don't completely get the motivation behind "simplifying" the hyperbolic layer. In the introduction, it is claimed that "the map defined in Chen et al. (2022) can be thought of a map from Euclidean space to Euclidean space". I guess that there are good reasons (accuracy, consistence?) for Chen et al. to define a map from the hyperbolic space to the hyperbolic space and to use Riemannian gradient descent.

The paper shows significant results of Mixed Geometry neural networks compared to Euclidean neural networks. However, as the authors mention in Section 5, it may be more significant to compare the performance with hyperbolic neural networks introduced in Chen et al. (2022) to see if the "simplification" of the layer has an impact on classification results.

**Submission Track:**

Extended Abstract (4 Page)

---

### Official Review · Reviewer_3Scy · 2022-10-19
**Evaluation**

**Confidence:** 3
**Soundness:** 3
**Presentation:** 2
**Contribution:** 3
**Overall Rating:** 5

**Summary:**

I think the logic of this work is straightforward and technically sound: 1) Hyperbolic graph neural network (HGNN) has achieved great success. 2) HGNN has the same issue as it implements most operations through differential and exponential maps in the tangent space. These operations are slow and create numerical instability. 3) Recently, the Lorentz model was introduced to HNN, which realizes the operations in the hyperbolic space via Lorentz transformations and resolves the issues mentioned earlier. 4) Thus, the authors propose to extend this development to the HGNN and show the benefit. This logic is fairly reasonable, and I tend to accept it.

**Questions:**

I have already mentioned earlier.

**Limitations:**

I have already mentioned earlier.

**Recommended Decision:**

3: Accept

**Relevance:**

3: Solid fit

**Strengths And Weaknesses:**

As I mentioned earlier, I would use this chance to discuss the weakness to help the authors polish the manuscript.

1) While using hyperbolic space to embed a tree structure in the representation space is relatively straightforward, the biggest question I have about these fully Hyperbolic networks is that the motivation for doing so is not clear enough. Why do we need many hyperbolic layers? Hierarchy of hierarchy? I would highly appreciate it if the authors could provide a stronger intuition, ideally with a simple example to demonstrate what is going on besides the dry benchmark numbers.

2) The presentation shall be improved.
Section 1, "Given a manifold M whose entire .. from M to itself." is a little confusing.
Section 2 - Fully Hyperbolic Network part is very disconnected from the following presentation. The author only says Both Lorentz boost and Lorentz rotation can be adopted as the basis for building fully hyperbolic neural networks, but shouldn't the author describe how these operations are used in F-HGNN to make the work more self-contained?
Section 3, $z_0 = .. \hat{x}_i^2$, small notation issue.

3) The numerical results are hard to evaluate. The authors should describe how significant these results are. Why do the experiments use only ten epochs? Where does the improvement come from? Can a reader trust these results are solid improvements due to F-HGNN, or are these results due to an under-optimized baseline? The authors shall make a significant effort in the future to make sure the improvements are solid.

**Submission Track:**

Extended Abstract (4 Page)

---

### Decision · Program_Chairs · 2022-10-21

Accept (Poster)